# On the Development of Phenol-Formaldehyde Resins Using a New Type of Lignin Extracted from Pine Wood with a Levulinic-Acid Based Solvent

**DOI:** 10.3390/molecules27092825

**Published:** 2022-04-29

**Authors:** Elodie Melro, Filipe E. Antunes, Artur J. M. Valente, Hugo Duarte, Anabela Romano, Bruno Medronho

**Affiliations:** 1Department of Chemistry, CQC, University of Coimbra, Rua Larga, 3004-535 Coimbra, Portugal; fcea@qui.uc.pt (F.E.A.); avalente@ci.uc.pt (A.J.M.V.); 2Science351—Disruptive & Sustainable R&D Innovations, Instituto Pedro Nunes, Ed. C, 3030-199 Coimbra, Portugal; 3MED—Mediterranean Institute for Agriculture, Environment and Development, Faculdade de Ciências e Tecnologia, Campus de Gambelas, Universidade do Algarve, Ed. 8, 8005-139 Faro, Portugal; hmduarte@ualg.pt (H.D.); aromano@ualg.pt (A.R.); bfmedronho@ualg.pt (B.M.); 4FSCN, Surface and Colloid Engineering, Mid Sweden University, SE-851 70 Sundsvall, Sweden

**Keywords:** bio-resins, phenolic resins, levulinic acid, lignin, lignocellulosic waste, resole resins

## Abstract

Resole resins have many applications, especially for foam production. However, the use of phenol, a key ingredient in resoles, has serious environmental and economic disadvantages. In this work, lignin extracted from pine wood using a “green” solvent, levulinic acid, was used to partially replace the non-sustainable phenol. The physicochemical properties of this novel resin were compared with resins composed of different types of commercial lignins. All resins were optimized to keep their free formaldehyde content below 1 wt%, by carefully adjusting the pH of the mixture. Substitution of phenol with lignin generally increases the viscosity of the resins, which is further increased with the lignin mass fraction. The addition of lignin decreases the kinetics of gelification of the resin. The type and amount of lignin also affect the thermal stability of the resins. It was possible to obtain resins with higher thermal stability than the standard phenol-formaldehyde resins without lignin. This work provides new insights regarding the development of lignin-based resoles as a very promising sustainable alternative to petrol-based resins.

## 1. Introduction

Phenolic resins are commonly obtained by the reaction of mono-hydroxybenzenes, especially phenol, and aldehydes, typically formaldehyde. Two major types of resins can be obtained: (1) resole resins, using a molar excess of formaldehyde in relation to phenol, under basic conditions, and (2) novalac resins, using a molar excess of phenol in relation to formaldehyde, under acidic conditions [1]. 

Phenolic resins have wide industrial interest, due to their superior mechanical strength, dimensional stability, high resistance against heat and flame, low smoke evolution upon incineration, and insolubility in water and several acids [2]. The choice between resole or novalac resins depends on the final application. For instance, to produce phenolic foams, resole resins are preferred [3]. Furthermore, this type of resin is also suitable for composites, reinforcing several materials [4,5,6,7,8]; in mineral wool/glass insulations, acting as a binder and making the insulations reliable and efficient [9]; in laminates, [10]; for curing epoxy and polyester coatings [11]; and as adhesives for the manufacture of plywood, particleboard, and oriented strand boards [12].

Phenol is a petroleum-derivative and, thus, its price is dependent on the petroleum quotation. Consequently, its price has been steadily increasing in the last years and this trend is not expected to change in the near future. Moreover, phenol is non-renewable, which leads the search for sustainable alternatives to replace it, bringing environmental and economic advantages. 

In this context, natural resources have been used to substitute phenol in phenol-formaldehyde resins, such as lignin, tannins, cardanol, palm oil, and coconut shell tar [13]. Among them, lignin is particularly relevant, due to its vast abundance (second most profuse biopolymer on earth), to produce bio-phenol formaldehyde (BPF) resins. Lignin is a complex and amorphous biopolyphenol, synthesized from the polymerization of three phenylpropane monomers: coumaryl, coniferyl, and sinapyl alcohols [14]. The rich aromatic structure of lignin makes it a very appealing sustainable alternative to replace petrol-derived phenol [15]. 

Different types of lignin have already been reported for the partially substitution of phenol, namely kraft lignin [16,17,18,19], soda lignin [20,21], sodium lignosulfonate [18], organosolv lignin [22,23], lignin bio-oil [24], enzymatic hydrolyzed lignin [25], and acid-insoluble lignin [26]. The modification of lignin, via demethylation [27,28] or depolymerization [29], has also been suggested to enhance lignin reactivity. The effect of the lignin type on the resole adhesive properties has been evaluated by Ghorbani et al. [20]. Nonetheless, the synthesis conditions were not optimized, like other published works using only one type of resin, which can result in resins with unacceptably high content of free formaldehyde. In the present work, this issue has been addressed and all formulations were optimized regarding the free formaldehyde concentration. Besides, the rheological behavior, rarely considered, was analyzed. Moreover, after the successful lignin dissolution in levulinic acid, LA [30], the solvent has been improved and here used to extract a new type of lignin from pine wood (henceforth called lignin-LA), which was then included for the first time in resole resin synthesis and its performance compared with different types of commercial lignins (i.e., dealkaline lignin, alkaline lignin, and lignosulfonate). 

## 2. Results and Discussion

The industrial requirements dictate that the free formaldehyde content of resole resins must be below 1 wt% [23]. Therefore, the synthesis protocol of all resins was optimized to accomplish this condition. The pH, solid content, free formaldehyde concentration, and chemically unbound lignin are parameters of interest, which are summarized in Table 1. For phenol-based resins, solid content of 49.1 (±0.7) wt%, and a free formaldehyde concentration of 0.26 (±0.02) wt% were obtained. On the other hand, when replacing 30 wt% phenol with lignin, using the same amount of NaOH, the free formaldehyde concentration is higher than without lignin, due to the lower reactiveness of lignin when compared to phenol [25]. The values are under 1 wt% for all commercial lignins, but for lignin-LA, the free formaldehyde content exceeds the critical value (1.84 (±0.03) wt%). This problem was tackled by increasing the NaOH concentration during the resole synthesis. The increase from 2 to 3 wt% NaOH decreases the free formaldehyde content (0.70 (±0.05) wt%). This optimization increases the pH of the reaction, but decreases the chemically unbound lignin, from 2.24 (±0.08) wt% to 1.1 (±0.2) wt%. The lowest values of free formaldehyde concentration and chemically unbound lignin are obtained using alkaline lignin. The solid content of the resole resins is observed to be ca. 50–53 wt%. 

For 50 wt% replacement of phenol by lignin, the lignosulfonate and alkaline lignins required only 2 wt% NaOH to present a free formaldehyde content below 1 wt.%. For the BPF systems composed of dealkaline lignin and lignin-LA, the pH was further adjusted; in the former case, the increase from 2 to 3 wt% NaOH results in a decrease in the free formaldehyde content from 1.27 (±0.03) wt% to 0.65 (±0.04) wt%, while in the latter case, adding 4 wt% NaOH decreases the free formaldehyde concentration to 0.77 (±0.08) wt%. It is interesting to note that the solid content of the resins with 50 wt% lignin is higher than 30 wt% lignin. Most likely, this observation is due to the high number of active sites for the higher concentration of lignin, which enhances crosslink networks.

After the synthesis of BPF resole resins, their viscosity over time was recorded. In Figure 1, the viscosity profiles of the different BPF fresh resins with 30 wt% replacement of phenol by lignin (and one resole without lignin for comparison) are shown. Except for BPF resin with lignosulfonate, the remaining resins experience an increase in viscosity with aging. Therefore, resins were stored at 4 °C after synthesis (viscosity remains constant at low temperatures suggesting slow kinetics of curing), and only removed when needed for further analysis. 

The viscoelastic behavior of the novel BPF resin with lignin-LA was accessed by amplitude sweep (Figure 2) and frequency sweep (Figure 3) assays. As can be observed in Figure 2, the resole without lignin displays a dominant liquid character (G’’, loss modulus, higher than G’, storage modulus). It is also possible to observe that the amplitude between these two components increases as the shear strain increases. For the BPF resin, with 50 wt% lignin-LA, the profile is similar when using 2 wt% NaOH. When doubling the NaOH concentration (to reach a resin with a free formaldehyde concentration below 1 wt%), both G’ and G’’ moduli are shifted to higher values, most likely due to an increase in formaldehyde and lignin that can now react under more suitable conditions (higher pH). Nevertheless, regardless of the NaOH concentration, the resins present an overall dominant liquid-like behavior with G’’ > G’ for all shear strains analyzed. Such behavior was generally observed for the other BPF resins with commercial lignins (Appendix A).

After the amplitude sweep measurements, constant shear stress of 5 Pa was selected (within the viscoelastic linear regime) to perform the frequency sweep tests. The resins without lignin show a typical viscoelastic behavior with G’’ > G’ at low frequencies and G’ > G’’ at higher frequencies (Figure 3). The crossover between both moduli determines the main relation time of the system. For the novel BPF resin with 50 wt% lignin-LA, the moduli crossover occurs at high frequencies for the 2 wt% NaOH which suggests that the liquid-like behavior is enhanced. This is further boosted for the 4 wt% NaOH formulation, where no G’ and G’’ crossover is observed within the frequency range tested, but both moduli are shifted to higher values. Similar behavior was observed for the BPF resins with the commercial lignins (Appendix A), with the exception of BPF resins with lignosulfonate, where the behavior is very similar, regardless of using 30 and 50 wt% lignosulfonate.

For an easier analysis of the rheological features, the complex viscosity, at 5 Pa and 1 Hz, was selected and compared for all resins represented in Figure 4a. The resin without lignin presents the lowest viscosity (0.073 Pa·s). The substitution of phenol by lignin increases the viscosity of the BPF resins. Except for lignosulfonate, where the repulsion among sulfonate groups may affect the interactions among polymers, the larger the amount of lignin, the higher the viscosity. The G’ and G’’ are represented in Figure 4b. As previously discussed, most resins have a dominant liquid-like character. This trend is maintained even for resins with higher viscosity, thus suggesting that no permanent physical or chemical networks have been established in the freshly prepared resins [31]. Data also indicate that upon the increase of NaOH concentration, the elastic behavior becomes more dominant (i.e., G’ > G’’), and the resins may present, in some cases, a solid-like character. This can suggest an increase in crosslinking reactions with increasing the alkalinity of the system (Appendix A).

The gel point of the uncured resins can be determined following the G’ and G’’ over time (Figure 5). It is important to note that the term “gel” is here used to represent the changes from a liquid to a rubbery state, with possible vitrification/solidification (transition from a rubbery to a glassy state). At 100 °C, the gel point of the standard phenolic resin without lignin occurs after 2106 s (i.e., 35.1 min). Remarkably, the viscoelastic parameters increase 5–6 orders of magnitude during this transition. When replacing phenol with lignin-LA, the kinetics of gelation increases considerably: the gel point for the novel BPF lignin-LA based resin was obtained at 1787 s (29.8 min), using 2 wt% NaOH, while this value drastically decreases to 380 s (6.3 min) for a NaOH content of 4 wt%. Such an enhancement of the gelation kinetics via replacing phenol by lignin-LA and further changing the pH (NaOH) might be very valuable in several applications regarding energy saving and productivity improvement and efficiency [32]. 

The molecular features of the BPF resins were further characterized by FTIR. As it can be observed, the spectra of the phenolic resin without lignin (Figure 6) is very similar to BPF resins with 30 wt% lignin (Appendix A), suggesting that, generically, all resoles have a similar molecular structure. An intense band is observed at 3298 cm^−1^ (#1), due to the stretching vibration of the hydroxyl groups. Small peaks at 2954 (#2) and 2842 cm^−1^ (#3) are assigned to the asymmetric and symmetric CH_2_ stretching vibrations modes, respectively. The region between 1638 and 1476 cm^−1^ (#4) shows different peaks related to the aromatic ring and the vibrations at 1455 (#5) and 1370 cm^−1^ (#6) are assigned to the methylene bending vibrations [33]. The band at 1234 cm^−1^ (#7) is characteristic of the C-O stretching and OH-deformation, while the bands at 1151 (#8) and 1110 cm^−1^ (#9) are characteristic of the aromatic C-H in-plane deformation [34]. The intense band at 1014 cm^−1^ (#10) is characteristic of C-O stretching. Between 890 to 686 cm^−1^, small peaks, related to the aromatic C-H out-of-plane deformations, are observed. The different peaks suggest different substitutions in the aromatic structure; the peak at 686 cm^−1^ (#14) indicates the presence of a phenol mono-substituted ring, while the peaks at 756 (#13), 830 (#12), and 890 cm^−1^ (#11), suggest the presence of di-, tri-, and tetra-substituted aromatic rings, respectively [35]. The novel BPF resin with lignin-LA is also present in Figure 6. The system with 2 wt% NaOH stands out regarding the intensity of the bands at 1595 cm^−1^ (C=C aromatic ring), 1021 cm^−1^ (-C-OH), and 754 cm^−1^ (CH aromatic, out-of-plane), and 692 cm^−1^ (ring bend), characteristic of phenol [36]. These peaks decrease significantly when the NaOH content is increased, due to the enhanced reactivity between the free phenol, or lignin, and formaldehyde. The main spectral differences among the 50 wt% lignin-substituted BPF resins rely on slight changes in the region of vibrations of the aromatic ring (1638–1476 cm^−1^) due to the high amount and different types of aromatic moieties (Appendix A).

The thermal stability of the phenolic resins was evaluated by TGA analysis, performed under a nitrogen atmosphere. The thermograms and respective first derivative curves (DTG) are represented in Figure 7. The main thermal events are summarized in Table 2. Typically, the thermal degradation of phenolic resins occurs in three principal steps, denoted as post-curing, thermal reforming, and ring stripping [37]. Between 100 °C and 250 °C, the first thermal event takes place, and its due to the evaporation of water generated by the condensation reactions of the methylol groups. The second thermal degradation occurs between 250 °C and 430 °C and it due to the loss of water formed by the condensation reaction among the methylene and phenolic hydroxyls and between the functional hydroxyl groups. The third thermal degradation occurs between 430 °C and 600 °C, and it is due to the elimination of carbon monoxide and methane formed by the degradation of the methylene bridge [28,37]. As it can be observed, the addition of alkaline lignin and lignosulfonate increases the T_max_ of the BPF resins in the initial stage, while the second thermal event is characterized by a global decrease of T_max_ for all lignins and for both concentrations (i.e., 30 wt% and 50 wt%). In the third thermal event, the addition of 30 wt% of commercial lignins (i.e., dealkaline lignin, alkaline lignin, or lignosulfonate) increases the thermal stability of the resin, compared to the resole without lignin, due to visible increase of T_max_ and weight residue at 600 °C. The novel BPF resin with lignin-LA shows a quite acceptable and comparable thermal stability to the commercial counterparts. Overall, these results suggest that the introduction of certain amounts of lignin in the resole formulation may increase its polymerization degree and concomitant thermal stability. Such behavior dependence on lignin concentration finds support in the rheological performance.

## 3. Materials and Methods

### 3.1. Materials

Phenol, hydroxylamine hydrochloride, and commercial lignins (i.e., dealkaline, alkaline, and lignosulfonate) were purchased from Tokyo Chemical Industry. Sodium hydroxide was acquired from José Manuel Gomes dos Santos, Lda. (Porto, Portugal), and formaldehyde (37 wt%) was obtained from Cham-Lab NV. Levulinic acid (98 wt%) was purchased from Sigma-Aldrich (Algés, Portugal), hydrochloric acid (37 wt%) was acquired from Fisher Scientific, and isopropanol obtained from Quimijuno (Coimbra, Portugal). Maritime pine (*Pinus pinaster* Ait.) sawdust was a gift from Valco–Madeiras e Derivados, S.A. (Leiria, Portugal).

### 3.2. Extraction of Lignin from Pine Wood Sawdust

The extraction process employed is similar to our previous work [38,39]. In brief, the pine sawdust was suspended in levulinic acid with 0.1 M HCl as the catalyst, and a solid-to-liquid ratio of 1:10 (g/mL). The mixture was kept under reflux in a paraffin bath at 140 °C for 2 h. The liquid fraction was then separated by vacuum filtration and 500 mL of deionized water was added to trigger the precipitation of lignin. Then, centrifugation was performed, and the solid material was dried in a freeze dryer. The extracted lignin is surprisingly rich in carbonyl groups in comparison with its commercial counterparts. For sake of simplicity, this novel type of lignin will be henceforth denominated as “lignin-LA”.

### 3.3. Preparation of Lignin-Based Resoles

The resoles were synthesized in a three-neck round-bottom flask equipped with a thermometer and a reflux condenser. The phenol-formaldehyde ratio was set to 1:1.80 for pure phenolic resins, and 1:1.30 and 1:1.00 for the bio-phenol formaldehyde (BPF) resins, replacing 30 and 50 wt% phenol with lignin, respectively [16].

In a typical reaction and considering the replacement of 30 wt% phenol, 6 g lignin (i.e., dealkaline, alkaline, lignosulfonate, or lignin-LA) were mixed with 14 g phenol, 4.1–5 g distilled water, and 1.8–2.7 g sodium hydroxide solution (50 wt%) in the flask and heated to 86 °C, during ca. 1 h, under magnetic stirring. Then, 22.4 g formaldehyde solution (37 wt%) was added to the flask and the reaction was maintained at the same temperature for 2 h extra. After this, the obtained resole was cooled in a water bath. The procedure was the same when replacing 50 wt% phenol with lignin. In this case, the amounts of the different compounds were 10 g phenol, 10 g lignin, 3.2–5 g distilled water, 1.8–3.6 g sodium hydroxide solution (50 wt%), and 17.3 g formaldehyde solution (37 wt%). For resole without lignin, 20 g phenol and 31.2 g formaldehyde solution (37 wt%) were used following the procedure described above. Despite these being the standard conditions used, to achieve a free formaldehyde concentration below 1 wt%, changes in the formulation were occasionally required, which mainly implied an increment in the NaOH concentration and strict pH control.

### 3.4. Characterization of the Resoles

The pH values of the resins were measured using a Jenway 924 005 C07 pH conjugated electrode coupled to an Inolab pH meter Level 1 (WTW).

The solid content was determined following the ASTM standard D4426-01 [40]. In brief, ca. 1 g of resin was weighed before and after being oven-dried at 125 °C for 2 h.

The free formaldehyde content was determined by the ISO 9397 method (hydroxylamine hydrochloride method) [41]. In a typical assay, ca. 1 g of resin was dissolved in an isopropanol-water-mixture (3:1). The pH was adjusted to 3.5 using 0.1 or 1 M hydrochloric acid and 25 mL of hydroxylamine hydrochloride solution (10 wt%) was added and the mixture was maintained under continuous stirring for 10 min. Finally, the solution was titrated to pH 3.5 using 0.1 M NaOH (aq.). The free formaldehyde was estimated following Equation (1).
Free formaldehyde (wt%) = 3 × NaOH concentration (M) × (V (mL) − V_0_ (mL))/m_sample_ (g),(1)
where V represented the titration volume and V_0_ in the volume from the blank titration. 

The chemically unbound lignin content was accessed following the method proposed by Zhou et al. [42]. In brief, 10 mL of NaOH (aq.) (5 wt%) were added to ca. 0.1 g cured resin. The mixture is then maintained at 50 °C for 24 h. The unbound lignin amount was estimated calorimetrically in a UV-Vis spectrophotometer (Shimadzu UV-2450, Shimadzu Corporation, Tokyo, Japan). Before measuring the samples, a calibration with each lignin was performed (concentration vs. absorption). For this, the wavelength of maximum absorption was determined for each lignin (i.e., dealkaline lignin: 286 nm, alkaline lignin: 296 nm, lignosulfonate: 286 nm, and lignin-LA: 299 nm).

The rheological measurements were carried out on a Kinexus Lab+ (Netzsch) at Science351, using a plate-plate geometry (20 mm, 0.5 mm gap). Non-linear (rotational) and linear (oscillatory) tests were performed using a peltier unit to accurately control de temperature (20 ± 0.1 °C). For the viscosity as a function of time assays, the rheological properties were accessed by placing a small amount of each freshly synthesized resin (ca. 1 g) in the rheometer and recording the viscosity over time, at 20 °C. The oscillatory measurements were performed using the resin stored at 4 °C. The complex viscosity, η* (complex modulus, G*, divided by angular frequency, ω [43]) was automatically calculated by the software of the rheometer.

Infrared spectra were recorded with a Thermo Nicolet 380 FTIR apparatus (Thermo Scientific, Waltham, MA, USA) equipped with a Smart Orbit Diamond ATR system. The spectra were measured from 4000 to 400 cm^−1^, in the absorbance mode, with a total of 68 scans and a resolution of 8 cm^−1^. Background spectra were collected before every assay. 

The thermal analysis of cured resins was evaluated using a thermogravimetric analyzer, TG 209 F Tarsus (Netzsch Instruments, Selb, Germany). In brief, ca. 3 mg of cured resin was heated from 30 °C to 600 °C at a heating rate of 10 °C.min^−1^ under N_2_ atmosphere (flow rate of 50 mL·min^−1^).

## 4. Conclusions

In this work, we have successfully reported the partial substitution of non-sustainable phenol by lignin, a bio-based polyphenol, in the development of BPF resoles. More importantly, novel resoles containing lignin extracted with a levulinic acid-based solvent were successfully developed and optimized. It was demonstrated that by precisely tuning the pH of the medium, via changing the NaOH amount, it is possible to achieve BPF resins with low free formaldehyde content and low chemically unbound lignin, thus fulfilling the industrial requirements. Overall, and regardless of the lignin type used, the systems present a dominant liquid-like behavior at room temperature. The kinetics of gelation is observed to enhance significantly for the resins with lignin, being the gelation point extremely dependent on the free formaldehyde and lignin unbound in the resin. The molecular insight from FTIR and the overall thermal behavior of resoles demonstrate that the novel BPF resins with the lignin extracted using a sustainable process (levulinic acid-based solvent) have a suitable performance and is similar to the systems developed with commercial lignins. Therefore, this work paves the way towards the use of a novel type of lignin, obtained with a renewable process, in the development of novel sustainable resoles that can efficiently compete and even replace phenol driven from non-sustainable sources. This also opens the possibility of implementing this novel type of lignin and the resole formulation here optimized in the development of other bio-based materials, such as phenolic foams.

## Figures and Tables

**Figure 1 molecules-27-02825-f001:**
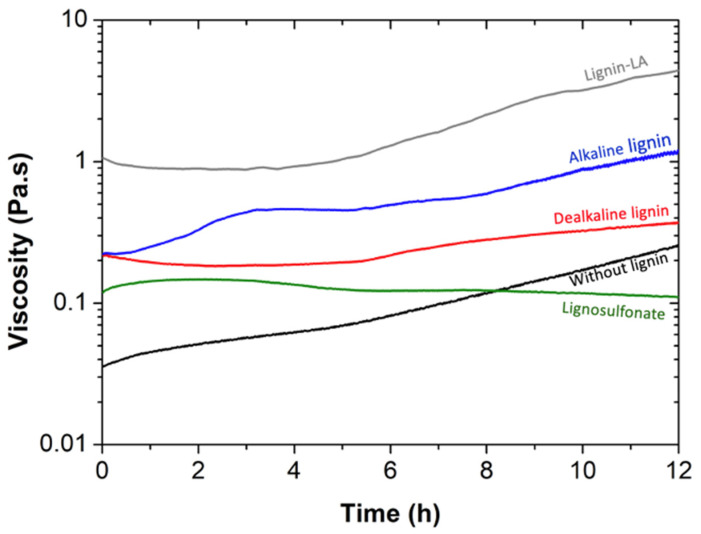
Viscosity as a function of time for resin without lignin (black), and BPF resins with 30 wt% dealkaline lignin (red), alkaline lignin (blue), lignosulfonate (green), and lignin-LA (gray). All the resins were produced using 2 wt% NaOH. The shear stress and temperature were kept constant at 10 Pa and 20 °C, respectively.

**Figure 2 molecules-27-02825-f002:**
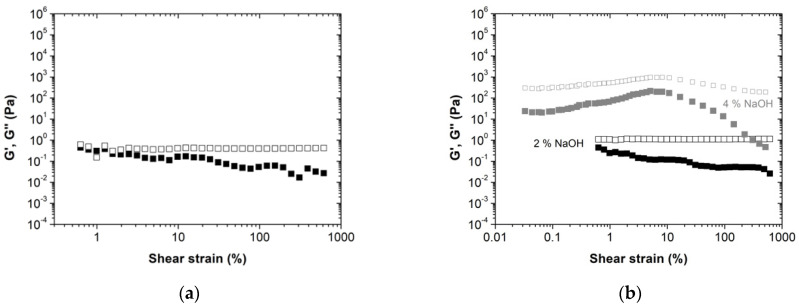
Strain amplitude sweep tests of resole resins: (**a**) without lignin; (**b**) with 50 wt% lignin-LA (right), at 1 Hz and 20 °C. The storage modulus, G’, is represented by full squares, while the loss modulus, G’’, is represented by the empty squares. The different colors of the symbols indicate the different NaOH concentrations: 2 wt%—black, 4 wt%—gray.

**Figure 3 molecules-27-02825-f003:**
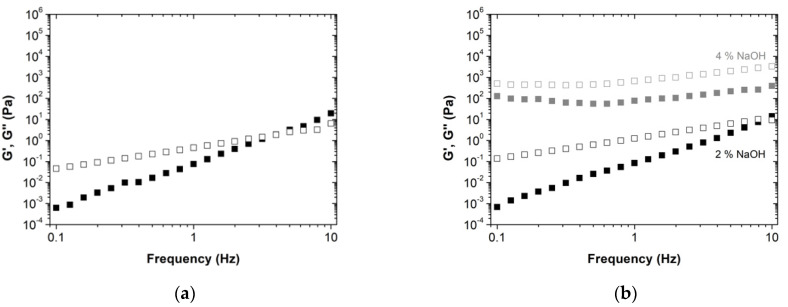
Frequency sweep tests of resole resins: (**a**) without lignin; (**b**) with 50 wt% lignin-LA (right), at 5 Pa and 20 °C. The storage modulus, G’, is represented by full squares, while the loss modulus, G’’, is represented by the empty squares. The different colors of the symbols indicate the different NaOH concentrations: 2 wt%—black, 4 wt%—gray.

**Figure 4 molecules-27-02825-f004:**
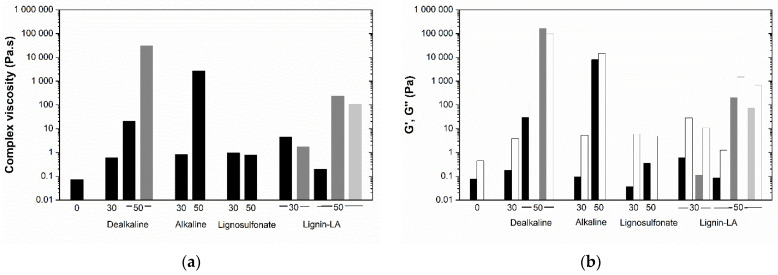
(**a**) Complex viscosity of resins; (**b**) G’ (full bars) and G’’ (empty bars) of resins (5 Pa, 1 Hz, 20 °C) with 0, 30 and 50 wt% of lignin, prepared with 2 wt% (black), 3 wt% (gray), and 4 wt% NaOH (light gray).

**Figure 5 molecules-27-02825-f005:**
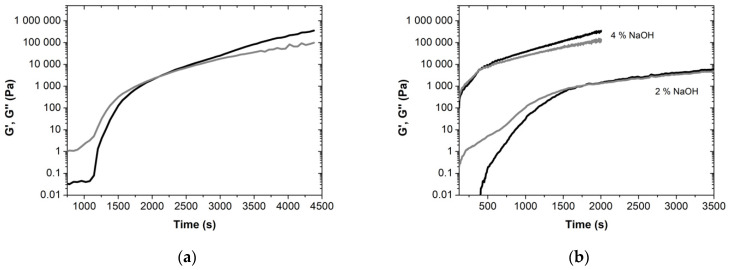
Kinetics of gelation of resole resins (**a**) without lignin; (**b**) with 50 wt% lignin-LA. The G’ (black) and G’’ (gray) are represented as a function of time, at 100 °C.

**Figure 6 molecules-27-02825-f006:**
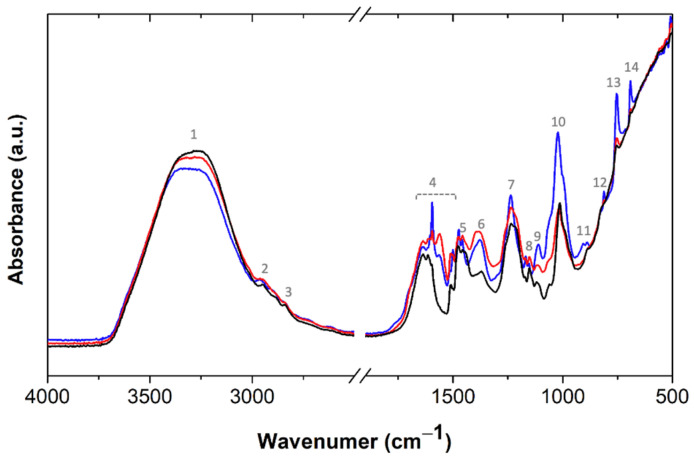
Normalized FTIR spectra of phenolic resins without lignin (black) and with 50 wt% lignin-LA, with 2 wt% NaOH (blue) and 4 wt% NaOH (red). The peaks of interest are highlighted, and their assignment is discussed in the text.

**Figure 7 molecules-27-02825-f007:**
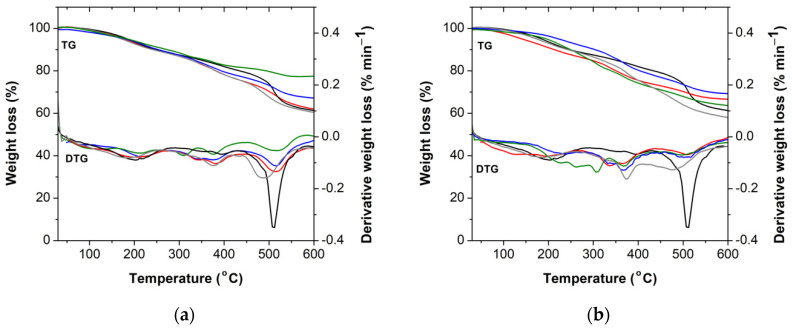
TG and DTG curves of resin without lignin (black), (**a**) with 30 wt% lignin; (**b**) with 50 wt% lignin. The used lignins were: dealkaline lignin (red), alkaline lignin (blue), lignosulfonate (green), and lignin-LA (gray). Note that all resins are optimized formulations (i.e., free formaldehyde content < 1 wt%).

**Table 1 molecules-27-02825-t001:** Physicochemical properties of resole resins.

Resin	Type of Lignin	NaOH (wt%)	pH	Solid Content (wt%)	Free Formaldehyde Content (wt%)	Chemically Unbound Lignin (wt%)
PF		2	9.4	49.1 ± 0.7	0.26 ± 0.02	-
30 wt% BPF	Dealkaline	2	9.2	50.5 ± 0.1	0.75 ± 0.03	1.5 ± 0.1
	Alkaline	2	9.9	51.2 ± 0.3	0.44 ± 0.06	0.28 ± 0.02
	Lignosulfonate	2	9.6	50.1 ± 0.4	0.49 ± 0.04	1.6 ± 0.0
	Lignin-LA	2	8.4	52.1 ± 0.4	1.84 ± 0.03	2.2 ± 0.1
		3	9.4	52.8 ± 0.0	0.70 ± 0.05	1.1 ± 0.2
50 wt% BPF	Dealkaline	2	8.6	55.8 ± 0.8	1.27 ± 0.03	5.2 ± 0.6
		3	9.2	54.4 ± 0.4	0.65 ± 0.04	0.89 ± 0.15
	Alkaline	2	10.3	56.7 ± 0.3	0.51 ± 0.06	0.68 ± 0.03
	Lignosulfonate	2	9.5	53.3 ± 0.1	0.52 ± 0.02	4.4 ± 0.1
	Lignin-LA	2	7.4	48.2 ± 0.5	7.98 ± 0.05	21 ± 1
		3	8.3	57.7 ± 0.3	1.20 ± 0.02	3.1 ± 0.7
		4	9.6	59.8 ± 0.2	0.77 ± 0.08	3.8 ± 0.8

**Table 2 molecules-27-02825-t002:** Summary of the main thermal properties of BPF resins.

Resin		First Thermal Event T_max_ (°C)	Second Thermal Event T_max_ (°C)	Third Thermal Event T_max_ (°C)	Weight Residue at 600 °C (wt%)
Without lignin		202	396	509	61.3
Type of lignin	wt% Phenol replaced				
Dealkaline lignin	30	203	382	515	62.7
	50	197	367	512	66.7
Alkaline lignin	30	215	375	516	67.2
	50	233	367	507	69.2
Lignosulfonate	30	213	375	516	77.5
	50	224	369	496	63.6
Lignin-LA	30	186	375	489	60.3
	50	186	373	478	58.0

## Data Availability

The data presented in this study are available in this article.

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
