# Peer review of "On the Development of Phenol-Formaldehyde Resins Using a New Type of Lignin Extracted from Pine Wood with a Levulinic-Acid Based Solvent"

_molecules, 2022, doi:10.3390/molecules27092825_

Round 1
Reviewer 1 Report
This work deals with the development of phenol-formaldehyde resins using new type of lignin extracted from pine wood with a levulinic-acid based solvent. The study is interesting but could be improved.
-The line 87 to 89 should be rephrased
-The discussion is lacking ( they are similar studies and the originality of the work is not compared with the litterature)
-this work per say " Fernandes et al.2021. New deep eutectic solvent assisted extraction of highly pure lignin from maritime pine sawdust (Pinus pinaster Ait.), international Journal of Biological Macromolecules" deals with a similar research topic but was not mentionned in the work.
-The authors talk about a leuvilic-acid based solvent is it a deep eutectic solvent? if not more explaination should be putting into the material and methods.
-I think the authors should use more recent references.
In general, the work is interesting, and should be ready for publication if little modification were applied.
Reviewer 2 Report
Review Report
General concept comment
The article named as, On the development of phenol-formaldehyde resins using a new type of lignin extracted from pine wood with a levulinic-acid based solvent, mainly deals with the characterization of phenolic resole resin based on lignin extracted with levunilic acid based solvent. Their properties are compared with those of other resins synthetized with commercial lignins. In particular, the effect of NaOH concentration used for the synthesis of resin based on lignin extracted with levunilic acid on rheological, thermal and chemical properties has been discussed. This article highlighted the feasibility to replace petroleum based phenol with lignin fraction extracted using a “green solvent”. Thermal stability and the decrease in gelification kinetics are improved.
The manuscript is globally clear. The article was presented in a good-structured manner. Most of the cited references are recent (mostly within the last 10 years). The result and discussion section is clear. I have appreciated the crosslinks done between all the results of various analyses led. However, Material and methods section, especially the section 3.2 and 3.3, should be clarified. Indeed, a part of the novelty of this work concern the improvement on resoles properties thanks partially to the use of lignin extracted and dissolved by levulinic acid. The extraction process is not presented: you just mention “paper submitted”. It is not sufficient to understand the preparation of this lignin fraction. I will recommend you to give more detail about the extraction method (operating conditions: pressure, mass or molar ratio of reactants …). I have found previous works led by your research teams: Polymers, 13, 2021, 7, International Journal of Biological Macromolecules, 164, 2020, 3454-3461). You may mention the latter in this research work if the paper submitted is not still accepted.
Moreover, in the section 3.3, the preparation of lignin bases resoles should be more detailed. As mentioned in the introduction section, you change the weight ratio of phenol and the NaOH concentration. Why do you also modify the weight of formaldehyde in function of weight of lignin used? I am wondering if this modification in formaldehyde content could have an effect on the characteristic of resole obtained compared with phenol resin (thermal stability, rheological)?
A second part of novelty mentioned in the introduction concern the optimization of operating condition of polymerization to reduce the content of free formaldehyde content in the resin. In the introduction section, a more detailed state of art could be developed concerning this point.
- Specific comments
Line 41-48: this sentence is too long. The part concerning the industrial interest of resole resin could be reduced. You could focus on the effect of operating condition on resole synthesis, for example.
Line 49-50;: sentence is not clear. The value corresponds to the cost and the petroleum quotation corresponds to crude oil price?
Line 102: “It is interesting to note that the solid content of the resins with 50 wt% lignin is higher than 30 wt% lignin”. Can you propose an explanation or a hypothesisfor these results?
Table 1: uniformize the significant digits of the columns “Chemically unbound lignin (wt %)” and “solid content”
Line 127/ Figure 2: “an increase in formaldehyde and lignin that can now react under more suitable conditions (higher pH)”. If I understood, the higher value of pH promotes the reaction between formaldehyde and lignin. However, can you clarify the link between the large signal shift of G’ modulus beyond 10 % and the fact more formaldehyde and lignin reacted together.
Line 160: In material and method, could you mention how the complex viscosity was obtained? Contrary to dynamic viscosity, what does represent this parameter?
Line 169-170: “the elastic behavior becomes more dominant, and the resins may present, in some cases, a solid-like character”. What is the parameter(s) which permits to identify and suppose this solid state? Can you give more explanations for this result?
Equation 1: what are the units used for resolving this equation concerning volume, concentration and mass sample?
Line 292: you can add the range of NaOH concentration used in yours experiments (from 2 to 4 %).
Line 314: Concerning this analysis, what is the wavenumber used for this UV analysis?
Reviewer 3 Report
Accept after the language editting.
Author Response
We are grateful for the positive feedback on our manuscript. The paper has been revised by English native speakers and was also checked by the Grammarly software.